# Do Employee Stock Ownership Plans Affect Corporate Social Responsibility? Evidence from China

**DOI:** 10.3390/ijerph19031055

**Published:** 2022-01-18

**Authors:** Lei Zhou, Feng Wei, Yu Kong

**Affiliations:** 1School of Economics and Business Administration, Chongqing University, Chongqing 400030, China; zzhoulei@cqu.edu.cn; 2School of Public Affairs, Chongqing University, Chongqing 400030, China; kongyu@cqu.edu.cn

**Keywords:** employee stock ownership plans, CSR, ownership structure, firm size

## Abstract

Few studies have discussed the relationship between employee stock ownership plans (ESOPs) and corporate social responsibility (CSR). Using a sample of 895 A-share public firms in China, this research examines the effects of ESOPs on CSR, and the moderating effects of wedge structure and firm size on this relationship. This research mainly used the OLS model to test the research hypotheses, and all regressions were performed in Stata15. The results show that the ESOPs of Chinese public firms provide external economic incentives and internal psychological incentives for employees, increase their motivation to engage in CSR activities, and ultimately contribute to CSR. At the same time, this research finds that this relationship is stronger for firms without wedge structure and small firms. This research provides insights for understanding the relationship between ESOPs and CSR and has important managerial implications for firms to pay attention to the interests of employees to achieve sustainable development.

## 1. Introduction

Since its introduction to the United States in the twentieth century, employee stock ownership plans (ESOPs) have attracted widespread attention from practitioners and academics as an internal corporate incentive. With the continuous improvement and development of ESOPs in practice, many firms in emerging capital market have started to adopt ESOPs. In June 2014, the China Securities Regulatory Commission (CSRC) promulgated the ‘Guiding Opinions on the Implementation of Pilot ESOPs by listed firms’ (Guidance), which formally restarted ESOPs for listed firms. As of 31 December 2019, more than 430,000 ordinary employees in China have subscribed for shares in their companies through ESOPs. Despite the significant academic attention directed to the relationship between ESOPs and firm performance, empirical findings remain mixed [1,2]. Some studies have indicated that ESOPs may reduce agency conflicts and promote cooperation and mutual monitoring among co-workers [3,4,5]. Other studies have argued that ESOPs are a measure for management to entrench itself and reverse takeovers, leading to poor corporate governance [6,7].

As academic research on ESOPs has progressed, the literature has shifted from focusing on the relationship between ESOPs and financial performance to several other relevant topics in non-financial performance [8]. However, as an important indicator in terms of non-financial performance, corporate social responsibility (CSR) has received little attention. In recent decades, the resources invested by organizations in CSR activities have rapidly increased [9]. A growing number of studies investigate the factors affecting the effect of the organizations’ individual level on CSR, such as employees’ emotions [10], individual characteristics [11], family image [12] and individual identity [13]. Although some studies have noted the important role of employee motivation in employee CSR engagement [14], few studies have focused on the effect of equity incentives, a common and important incentive measure for employees, on CSR. To address these research gaps, we examine the links between ESOPs and CSR, and the role of wedge structure and firm size in moderating the relationship between ESOPs and CSR.

Employees, as the firm’s key stakeholders, play a crucial role in the development and implementation of CSR strategies [15,16]. They are mainly responsible for the fulfilment of corporate ethical behaviors in their daily work, and the achievement of a firm’s CSR goals largely depends on employees’ collaboration [17,18]. Employee participation in CSR activities frequently occurs in the form of corporate volunteer programs, in which employees offer their time and skills in service to the community. Employees have also made great contributions to corporate philanthropic donations, both outside the firm or inside the firm (e.g., employees and colleagues). In addition, employees play an important role in social services (e.g., product production and after-sales service) and environmental protection [19]. For example, Afsar et al. [20] found that environmental sustainability at the organizational level is largely shaped by and dependent on environmental behavior at the individual employee level. More importantly, some studies have suggested that the role of employee participation in CSR is not confined to the implementation of CSR activities as employees are also able to suggest CSR policies [21].

Using a sample of 895 A-share listed firms in China, this paper came to the strong and robust conclusion that ESOPs have a significantly positive effect on CSR. This finding complements the view that ESOPs motivate employees to not only improve corporate financial performance pay attention to the interests of the firm’s stakeholders. The positive relationship between ESOPs and CSR is also robust to a series of sensitivity tests. Additionally, we find that for firms without wedge structures and small firms, the positive relationship between ESOPs and CSR performance is stronger.

This paper makes the following main contributions. First, we offer a new perspective in the study of organizations’ individual level influencing factors of CSR. To the best of our knowledge, this is the first study to specifically examine the relationship between ESOPs and CSR. Previous studies on ESOPs have focused on corporate governance and financial performance [2,22], such as the effect of ESOPs on equity return [4], R&D investment [21] and information disclosure [2]. In particular, few studies have focused on the effect of ESOPs on CSR. Second, our study shows that the incentive effect of ESOPs depends on some specific conditions. We find that the wedge structure will weaken the incentive effect of ESOPs. At the same time, we find that the relationship between ESOPs and CSR is affected by firm size, and ESOPs are susceptible to free-riding problems in large firms. Our study provides insights into the environment in which ESOPs are implemented, where decentralized ownership structure and small firms may be the ideal choice to make ESOPs effective.

The remainder of this paper is structured as follows. The second part provides the institutional background and hypotheses. The third part presents the sample selection, data sources and sample characteristics. The fourth part provides the empirical analysis, including robustness and endogenous tests. The last section is dedicated to the discussion, implications and conclusion of the current survey.

## 2. Institutional Background and Research Hypotheses

### 2.1. Institutional Background

ESOPs in China have three main stages, as follows. First stage—China’s ESOPs actually began with internal employee share ownership as part of a state-owned enterprises (SOEs) reform. The Chinese government started the process of corporatization in 1992 that allowed SOEs to be privatized through share ownership and become an incorporated company. After SOEs become corporations, with the approval of regulatory authorities, SOEs can issue part of the equity to internal employees to improve their operating efficiency [23]. However, owing to the lack of effective supervision, a large number of irregularities exist in ESOPs at this stage, resulting in a significant loss of state-owned assets, and the Chinese government eventually terminated the internal employee share ownership policy in 1994. Since then, the internal employee share of Chinese listed firms has gradually decreased. After 2007, all internal employee shares were converted into tradable shares.

Second stage—China implemented the split-share reform in 2005, which was a milestone in the development of the Chinese capital market by increasing the liquidity and information content of the shares held by controlling shareholders; this better realizes the alignment of interests between controlling shareholders and minority shareholders [4]. To adapt to these changes and improve corporate governance and performance, Chinese listed firms were to provide better incentives to their employees. Therefore, the CSRC issued the ‘Regulation of Equity Incentive Plans (trial)’ that specified the general rules for firms to grant employee equity incentives, such as restricted stocks and stock options, in late 2005. The procedures for the implementation of equity incentive plans, information disclosure, regulatory procedures and penalties when violations occur are included in the rules. However, the main incentive objects of the equity incentive plans are the directors, supervisors and managers of listed firms, rather than the ordinary employees.

Third stage—In June 2014, the CSRC promulgated the ‘Guidance’, formally launching the ESOPs for listed firms. Although ESOPs and equity incentive plans are consistent in their goals of binding the interests of employees and shareholders, institutional designs have significant differences. For example, while equity incentive plans focus on motivating management, ESOPs are broadly distributed to employees. Equity incentive plans mainly include the granting of restricted stocks and stock options, and the management is accompanied by corresponding performance evaluation. Conversely, ESOPs are mainly administered through the listed company to repurchase the firm’s shares, as secondary market purchase and subscription of non-public shares, but without a related performance evaluation. In the former type of plans, provided that the management meets the performance standards, they can decide whether to drive the corresponding options. The shares held by employees in the ESOPs are generally managed centrally by an organization elected by employees, and the employees cannot sell their shares for a short period.

We were motivated to examine the relationship between ESOPs and CSR in China for the following reasons. First, Chinese listed firms have played a key role in the rapid growth of China’s economy. As of 2020, there were more than 4000 A-share listed firms in China, with a total market value of CNY 76 trillion and counting 133 Chinese firms in the Fortune Global 500. However, since the Chinese government started the process of corporatization in 1992, Chinese listed firms are relatively young at this stage, and many listed firms are facing weak corporate governance problems. To this end, China has learned from developed Western countries and introduced a series of reform policies including ESOPs aiming to improve the corporate governance level of Chinese listed firms. It is important to examine the effects of these policies, because these can provide important reference value for other emerging capital market countries such as China.

Second, in recent decades, excessive emphasis on economic growth has led many Chinese firms to ignore the fulfillment of CSR, and incidents such as Sanlu’s tainted milk powder and Changsheng’s fake vaccines have occurred. Although the Chinese government has introduced various regulations to encourage listed firms to fulfill their CSR, many listed firms still lack CSR. For example, only 1081 A-share listed firms issued independent CSR reports in 2019, and more than two-thirds of the listed firms did not disclose independent CSR reports. Among them, only 512 listed firms made voluntary disclosures, while other listed firms made mandatory disclosures in accordance with legal regulations. This reflects the inherent limitations of external institutional constraints such as government regulations, laws and enforcement mechanisms, which have driven us to consider whether the corporate internal governance such as employee incentives can affect the fulfillment of CSR.

### 2.2. Hypotheses Development

As a key group of a firm’s stakeholders, it is well known that employees play a key role in the development of a firm’s CSR strategies and activities [24,25,26]. Collier and Esteban [17] emphasized the dependence of organizations on employees’ responsiveness and participation in CSR. They argued that the effective implementation of CSR projects depends on the willingness of employees to collaborate. However, not all employees will be equally and actively engaged in organizational CSR goals. The financial transactions between employees and organizations are typically explicit contractual agreements from which employees can profit. In contrast, fulfilling CSR may exceed the scope of the normal economic contract between the employee and the organization and be considered a social contract, but the social contract does not have clearly defined responsibilities and obligations [27]. Hence, some employees may not be sufficiently motivated to engage in CSR. Rodrigo and Arenas [28] argue that organizations often have some ‘indifferent’ employees who are pragmatic, goal-oriented and personally indifferent to CSR engagement—even if they understand the role of CSR in the organization. Additionally, there are some ‘dissident’ employees who only regard work as an economic contract with no responsibility to a wider social role.

The principal–agent theory can also explain the absence of employees in CSR. CSR activities are a complex process with high investment, multiple levels and long cycles, and this require the engagement of employees from all departments. Agency theory argues that, in the corporate principal–agent relationship, the goals of agents (i.e., managers and employees) are often inconsistent with the interests of principals (i.e., shareholders) [29]. For example, prior literature has argued that managers and workers are natural allies against takeover threats because takeovers and subsequent mergers are often associated with layoffs [30,31]. For shareholders, CSR can give a firm good reputation and increase shareholder value. Previous literature has shown that CSR can give a firm good reputation and increase shareholder value because better CSR can reduce corporate risk [32], increase investment efficiency [33], improve corporate reputation [34] and ultimately increase the long-term firm value [35]. However, employees only receive a fixed salary for their work, do not share in the ownership of firm assets and residual earnings and do not enjoy the reputational and financial benefits of high CSR, and some even profit from activities that undermine the firm’s CSR (e.g., Sanlu’s tainted milk powder incident was caused by typical employees neglecting CSR in pursuit of private interests).

Collier and Esteban [17] believe that, to ensure that employees fulfil the requirements related to CSR goals, they should be given sufficient incentives. We posit that ESOPs motivate employees’ CSR engagement with at least the three following aspects. First, ESOPs can coordinate the interests of employees and shareholders and thus effectively mitigate the aforementioned agency conflicts [36]. These create an ecosystem of incentive contracts throughout the organization, thus aligning the interests of employees and owners [5]. By binding employee earnings to firm stock value, ESOPs can motivate employees to engage in CSR activities from external economic incentives. When employees hold firm shares, they will do their best to maintain and improve the firm reputation and image closely related to the firm stock value in the organizations’ daily activities. Previous literature has well documented that CSR is a key activity that affects firm reputation [33]. At the same time, some studies have found that there is also a positive relationship between CSR and shareholder value [37,38]. For example, Lins et al. [39] found that the stocks of higher CSR firms performed better than those of lower firms during the 2008–2009 financial crisis, and the firm-specific social capital established by engaging in CSR will be rewarded once society’s trust has been restored. This is also consistent with the results found in Nofsinger and Varma [40].

Second, in addition to external economic incentives, the internal psychological incentives of ESOPs can also encourage employees to engage in CSR. Before the implementation of ESOPs, the limited labor of employees paid to work was the optimal choice. When employees hold shares in the firm, they technically become owners of the firm, allowing their employees to feel a sense of ownership [41], which will greatly improve employees’ organizational identity and organizational commitment. For example, Chiu and Tsai [42] found that stock-based profit sharing has a positive effect on employees’ organizational citizenship behavior. Organizational identity and organizational commitment is an important motivation for employees to engage in CSR activities, which ensures that they are motivated to implement CSR practices. There exists a stream of literature that views underlying psychological outcomes such as organizational identity and organizational commitment to be the key factors affecting employees’ individual levels to carry out CSR activities [43]. For example, as suggested by Collier and Esteban [17], if organizational attributes are perceived as attractive by employees, they will identify strongly with the organization, and strong organizational identification may translate into cooperative and citizenship-type behaviors.

Finally, the above two kinds of incentives will increase mutual supervision among employees [44], reducing the behavior of employees seeking private interests from activities that damage organizational CSR. Hochberg and Lindsey [45] argued that the value of ESOPs as a group incentive plan is demonstrated by employees working together; this enhances cooperation among employees which can also lead to mutual monitoring among colleagues. Furthermore, Sesil et al. [46] argued that, owing to the complexity of the firm’s mission, shareholders may not have the adequate conditions to monitor the decisions of managers and employees. However, employees may be better qualified than shareholders to monitor the quality of each other’s contributions at work. Giving employees incentives for self-monitoring and peer monitoring through collective incentive schemes depending on performance may be a cost-effective alternative to formal monitoring (e.g., more supervisors). In particular, should employees collectively agree to exert this supplementary effort, the incentive to monitor and sanction their colleagues engaging in CSR will increase, because each employee’s actions affect organizational CSR goals [45]. Thus, although an employee could profit from actions that undermine the firm’s CSR, other responsible employees would not allow it, as this would compromise the firm’s value and reputation and thus affect their compensation. Based on this analysis, we propose the first research hypothesis as follows:

**Hypothesis** **1** **(H1).**
*Compared to firms without ESOPs, firms with ESOPs have higher CSR.*


ESOPs enable employees to become the owners of the firm and encourage them to participate in business decisions; however, the effectiveness of ESOPs depends on the corporate ownership structure [4]. La Porta et al. [47] found that, except for a few countries with developed capital markets, ownership concentration is a relatively common ownership structure worldwide, and firms often have the ultimate controller. In emerging capital markets such as China, the ultimate controller will control the firm through pyramidal shareholding and cross shareholding [48], where pyramidal shareholding separates control and cash flow rights [49]. Thus, as long as the controlling shareholder’s gain from using control to transfer the corporate resources is less than the loss suffered due to the existence of cash flow rights, the controlling shareholder has sufficient incentive to intervene in the interests of other shareholders.

When the degree of wedge structure is high in listed firms, the controlling shareholder intervenes in the interest of minority shareholders by diluting wages, delaying transactions and transferring assets, but other shareholders will not be able to perform effective supervision [50]. In particular, in many emerging capital markets, wedge structures may be associated with weaker oversight [48]. In such cases, the wedge structure can weaken the incentive effect of ESOPs because if the positive and governance effects of ESOPs limit the private interests of controlling shareholders, controlling shareholders may resist employee participation in corporate governance. Furthermore, as the wedge structure expands the controlling shareholders’ control, employees find it difficult to engage in corporate decision making in a real sense, and the equity interests of employees are also vulnerable to the infringement of the controlling shareholder. Based on this analysis, we propose the second hypothesis as follows:

**Hypothesis** **2** **(H2).**
*Wedge structure weakens the positive effect of ESOPs on CSR.*


As ESOPs in Chinese listed firms target the majority of employees, some studies have argued that free-riding problems hinder the effectiveness of ESOPs [51]. According to free-riding theory, collective action will benefit members who do not share the costs and risks. In profit-sharing or employee-owned firms, the marginal effort of any single employee will be shared by many other members. Thus, when firms have a large number of employees, individual employees may feel that their actions have little impact on the firm’s goals; therefore, they may be reluctant to perform tasks that require extra effort or sacrifice. This free-rider effect, often referred to as the 1/N effect, intensifies as the number of employees, N, increases [51]. Conversely, firms with a small number of employees are less likely to experience free-riding as corporate performance is more sensitive to the behavior of individual employees, and individual employees have a higher ability to influence firm value [52]. Related research has found that, in firms with a small number of employees, ESOPs provide stronger incentives for employees, so they are more motivated to increase firm performance [3,5,44]. We speculated that the free-rider effect persists in the relationship between ESOPs and CSR. Based on this analysis, the following hypothesis can be proposed. Figure 1 provides the theoretical framework of this paper.

**Hypothesis** **3** **(H3).**
*Firm size weakens the positive effect of ESOPs on CSR.*


## 3. Data and Research Design

### 3.1. Sample and Data Collection

We used a sample of 895 Chinese A-share listed firms from 2014 to 2018. The ESOP data were obtained from the WIND database, the corporate financial data and other data were from the CSMAR database, and the CSR data were from the China Research Data Service Platform (CNRDS) database. We further screened the sample in the following manner: we excluded firms in the financial industry owing to their unique disclosure requirements and accounting rules. We also excluded firms with more liabilities than assets (i.e., companies with an asset–liability ratio higher than 1) and firms with ST and *ST treatments (i.e., companies in trouble). In addition, we excluded firms with equity incentive plans to prevent the cross-over effects of equity incentive plans and ESOPs [51]. Ultimately, we obtained 4142 firm–year observations for 895 listed firms, among which 173 firms adopted ESOPs during the sample period.

### 3.2. Model

To examine the relationship between ESOPs and CSR, we formulated the following equation (please refer to Appendix A for detailed definitions of all the variables in Equation (1)):CSR_i,t_ = β_0_ + β_1_ESOP_i,t_ + αControl_i,t_ + ɛ(1)

Here, i and t represent the firm and time subscale indicators, respectively.

In CSR research, many studies have used the well-known KLD dataset. However, this dataset only covers US firms. The CSR data used in this study are from the China Research Data Service Platform (CNRDS) database. The design of the CNRDS database is with reference to the design of many well-known international CSR databases (e.g., the KLD database) and releases data depending on the content of the CSR report. The CNRDS database includes CSR data of all public firms that have disclosed their CSR reports on the Shanghai and Shenzhen stock exchanges from 2006, which makes us maximize the data coverage. We collected data from five CSR dimensions: philanthropy and volunteerism; CSR disclosure; employee relations; environmental responsibility; and products. Table 1 reports the definitions of different CSR dimension indicators. Subsequently, we calculated the CSR score based on five different CSR dimensions, and the secondary indicators for each CSR dimension are all dummy variables is 1 if the firm has an advantage on that secondary indicator and 0 otherwise.

First, based on Deng et al. [37] and Cheung et al. [53], we used the following formula to calculate our first dependent variable:(2)$$\mathrm{CSR}\text{\_}\mathrm{FRA}=\sum_{1}^{i}\frac{{\mathrm{score}}^{i}}{n^{i}},\;$$
where the score^i^ is the CSR score for dimension i and n^i^ is the total number of indicators of CSR in dimension i.

Second, we logged the total CSR score of five CSR dimensions (38 qualitative indicators across five CSR regions) to obtain our second dependent variable, CSR_LOG.

We used ESOP to measure whether firms adopt ESOPs. If firms adopt ESOPs in that year, ESOP is equal to 1; otherwise 0.

Control is the control variable in our paper. Following previous research [54,55], we selected variables that have a significant impact on CSR. We used the log of total assets to measure the firm size (SIZE) rather than the number of employees to measure firm size because total assets are an important factor affecting a firm’s CSR investment. The corporate IPO year to that measures the firm age (AGE), ratio of earnings to total assets to measure firm profitability (ROA), ratio of cash holdings to total assets to measure firm cash holdings (CASH), ratio of total liabilities to total assets to measure firm leverage (LEV), annual growth rate of firm assets (GROWN), annual growth rate of the corporate operating income (INCOME) and the ratio of management expenses to operating income to measure management expense ratio (MANAGE). In addition, we included the ratio of independent directors (INDRATE) and whether the chairman is also the CEO (DUALITY) as control variables (please refer to Appendix A for detailed definitions of control variables).

To examine the moderating effect of wedge structure on the relationship between ESOPs and CSR, we formulated the following equation:CSR_i,t_ = β_0_ + β_1_ESOP_i,t_ + β_2_WEDGE_i,t_ + β_3_ESOP∗WEDGE_i,t_ + αControl_i,t_ + ɛ(3)

WEDGE is a dummy variable: when the control and cash flow rights of the public firm are separated, WEDGE is equal to 1, and otherwise 0.

To examine the moderating effect of firm size on the relationship between ESOPs and CSR, we formulated the following equation:CSR_i,t_ = β_0_ + β_1_ESOP_i,t_ + β_2_FIRMSIZE_i,t_ + β_3_ESOP∗FIRMSIZE_i,t_ + αControl_i,t_ + ɛ(4)

FIMESIZE is a dummy variable: when firms with employee’s number less than or equal to the median of the same industry in that year, FIRMSIZE equals 1; otherwise 0.

All of the above equations use the OLS model for regression. To ensure the robustness of the results, we will use the alternative CSR measure and CSR lag term for robustness testing. At the same time, we also used the difference-in-differences model and Heckman two-stage model for endogeneity testing to ensure that our research does not suffer from serious endogeneity problems.

### 3.3. Descriptive Statistics

Table 2 panel B reports the descriptive statistics of the kay variables. The mean of ESOP is 0.123, indicating that only approximately 12% of the observations in our sample have ESOPs. The mean of CSR_FRA and CSR_LOG are 2.849 and 2.271, and the difference between the maximum and minimum values is significant; this suggests the prominent individual differences in the CSR of Chinese listed firms and the lack of strong CSR sense in many listed firms. The mean and median of AGE are 13.14 and 14, respectively, suggesting that the relatively young Chinese listed firms (as China has only opened its stock market since 1990). In addition, the mean of SIZE, ROA and LEV are 23.25, 0.041 and 0.482, which are similar findings to those of previous studies.

## 4. Empirical Results

### 4.1. Multivariate Regression Analysis

Table 3 presents the results of OLS estimations for the relationship between ESOPs and CSR. As shown in Model 1, ESOP has a positive and significant relationship with CSR (coefficient = 0.036 at *p* < 0.05), which provided strong support for Hypothesis 1. In Model 2, the interaction term between ESOP and WEDGE was included and its coefficient was negative and significant (coefficient = −0.005 at *p* < 0.01), which provided strong support for Hypothesis 2. In Model 3, the interaction term between ESOP and FIRMSIZE was included and its coefficient was negative and significant (coefficient = −0.057 at *p* < 0.1), which was consistent with Hypothesis 3. To achieve a more reliable conclusion, we used the log of the total CSR score of five CSR dimensions to measure our second dependent variable. The regression results of Models 4–6 are consistent with the aforementioned benchmark regression results, which proves that our research conclusions have good robustness.

### 4.2. The Sustained Impact of ESOPs CSR

To explore the impact of ESOP on CSR in the long term, we used T + 1 and T + 2 periods of CSR as the dependent variable in Table 4, respectively. Again, the coefficients of ESOP are significantly positive in both cases, and the corresponding values are 0.044 and 0.062, suggesting that the CSR of firms with ESOPs increases by 4.4% in year t + 1 and a 6.2% increase in year t + 2. Furthermore, we found that the interaction terms between ESOP and WEDGE, ESOP and FIRMSIZE are still negatively significant for CSR_FRAt + 1 and CSR_FRAt + 2, suggesting that the moderating effect of wedge structure and firm size on the relationship between ESOPs and CSR is a long-term existence. When we used CSR_LOGt + 1 and CSR_LOGt + 2 as the dependent variables, the regression results are consistent with the aforementioned benchmark regression results, which proves that our research conclusions have good robustness.

### 4.3. Endogeneity

We used the lag effect to verify the relationship between ESOPs and CSR, alleviating certain endogenous concerns. However, the relationship between ESOPs and CSR may encounter other endogenous problems. The causal relationship between ownership structure and firm characteristics is difficult to identify [56], and research on the relationship between ownership structure and CSR is susceptible to reverse causation. Determining whether ESOPs increase CSR or whether higher CSR firms are more concerned with the employee’s interests and therefore implement ESOPs is also a difficult task. Moreover, as the choice of ownership structure may not be random [57], our study may encounter the endogenous problems of sample self-selection. Hence, we followed the previous literature [58,59] and used the difference-in-differences model (DID) to address potential reverse causal endogeneity concerns, using Heckman two-stage regression mitigate sample selection bias endogeneity concerns. We discuss these analyses in detail below.

#### 4.3.1. Difference-In-Differences Model

To better establish the causal relationship between ESOPs and CSR and mitigate the biased estimation caused by missing variables that simultaneously influence firms with and without ESOPs, we followed previous studies [57,58] and employed the staggered DID method to estimate the difference in CSR before and after the ESOPs were initiated. Based on the exogenous event that led the CSRC to initiate the ESOPs, we examined the difference in CSR between the experimental group and the control group before and after the adoption of ESOPs. We specifically used the firms that adopted ESOPs during the entire sample period and the firms that never adopted ESOPs during the entire sample period for the DID model test. Firms with ESOPs were included in the ‘treatment group’ (TREAT = 1), and we identified a ‘control group’ (TREAT = 0) of firms without ESOPs for the entire sample period. The DID model is expressed as follows:CSR_i,t_ = β_0_ + β_1_AFTER∗TREAT_i,t_ + β2TREAT_i,t_ + αCONTROL_i,t_ + ɛ(5)

Here, AFTER is an indicator of the period after the adoption of ESOPs, which equals 1 if firm i adopts ESOPs in year t; otherwise, 0. Thus, the interaction of AFTER and TREAT (AFTER*TREAT) measures the absolute effect of ESOPs on CSR. Furthermore, CONTROL is the control variables that is the same as in the main regression. We also included annual and industry dummy variables to control for the effects of time trend factors, such as macro policies and industry factors. β1 measures the difference in CSR before and after the implementation of ESOPs. The results are reported in Table 5. The coefficients of AFTER*TREAT are significantly positive for CSR_FRA and CSR_LOG, suggesting that CSR will significantly increase for firms with ESOPs. In summary, the results suggest a positive causal effect of ESOPs on CSR.

#### 4.3.2. Heckman Two-Stage Regression

A firm’s choice of ESOPs may not be random, but rather determined by certain firm characteristic factors. To minimize concerns over sample selection bias, we adopted Heckman two-stage regression to examine the relationship between ESOPs and CSR. In columns (1) of Table 6, we report the Probit regression results of the first stage of the Heckman model where the dependent variable is whether the firm adopted ESOPs. In columns (2)–(7), we report the regression results of the second stage of the Heckman model, and the coefficients of EOSP are still significantly positive for CSR_FRA and CSR_LOG, which are consistent with the results of OLS. Moreover, the coefficient of the inverse Mills coefficient is not statistically significant, suggesting that there is no sample selection bias in our study.

## 5. Discussion

Our study contributes to the literature on factors affecting CSR. Specifically, our study contributes to the growing discussion on the effect of the organization’s individual level on CSR [10,11]. In contrast to previous studies pertaining to CSR in employee characteristics [60,61], our study mainly focused on the impact of employee incentives from ESOPs on CSR. We found that compared to firms without ESOPs, firms with ESOPs have higher CSR; this finding is consistent with the results of extant empirical studies [45,52,62], suggesting that through the binding of employee income and corporate interests, ESOPs promote the role of employees as owners and make employees pay more attention to activities related to long-term firm value.

Our study also attempted to extend the discussion on how ESOPs affect firm outcomes [3,63]. Prior studies have examined the effect of ESOPs on corporate financial performance, but few studies have focused on the effect of ESOPs on CSR. We argue that the economic and psychological incentives that ESOPs bring to employees can alleviate the agency conflict between employees and shareholders in fulfilling CSR. As previously stated, the long-term benefits of CSR for the business benefit employees who hold firm shares and organizations identify that bringing ESOPs to employees can encourage employees to be more willing to engage in social and environment-related activities carried out by the firm. Since employees are among the most important stakeholders of the firm, our results imply that a firm can balance the interests of key stakeholders through ESOPs. Rodrigo and Arenas [28] argued that organizations often have some ’indifferent’ and ’dissident’ employees who are pragmatic and economic contract-oriented, who are personally indifferent to CSR engagement. The economic incentives of ESOPs have the potential to change such employees’ CSR preferences, suggesting that ESOPs contribute to employees’ engagement in a firm’s sustainable development.

In addition, our study has some critical practical implications for the conditions under which ESOPs are implemented. First, our results show that a wedge structure weakens the positive effect of ESOPs on CSR. The excessive control rights of large shareholders may be detrimental to the incentive effect of ESOPs. On the contrary, the dispersed ownership structure allows employees who hold a small ratio share to play a role in the firm’s decision making. This is an important insight, particularly in emerging capital market countries such as China, where the ownership structure of listed companies is relatively concentrated [47,48]. Policy makers or listed firms should take some measures to protect the legal rights of employees who hold firm shares, to ensure that employees’ stock rights will not be infringed upon by large shareholders, thus increasing the enthusiasm of employees who hold shares to participate in the firm’s business decisions. Second, our results show that firm size weakens the positive relationship between ESOPs and CSR, and the ESOPs of large firms are more likely to suffer from free-rider problems. This finding is consistent with the empirical results of Hochberg and Lindsey and Kim and Ouimet [45,51]. Therefore, firms should control the scale of implementation of ESOPs to prevent the disadvantages of averaging. Overall, our study provides insights into the environment in which ESOPs are implemented, where decentralized ownership structure and small firms may be the ideal conditions for ESOPs.

## 6. Limitations and Future Research Directions

Although this paper provides some important insights, this paper has several limitations that should be addressed in future research. First, although we considered the role of economic incentives and psychological incentives in the relationship between ESOPs and CSR, our empirical research only focused on the direct effect of ESOPs on CSR. We thus suggest that future research may use mediating factors to explore the effect mechanism of ESOPs on CSR. Second, the CSR indicators in this study are measured by five CSR dimensions: philanthropy and volunteerism; CSR disclosure; employee relations; environmental responsibility; and products. We may have overlooked the preferences of employees regarding each of the different CSR dimensions. Future research may consider examining the effect of ESOPs on different dimensions of CSR. Finally, the study sample was only limited to China, and the relationship among ESOPs, wedge structure and CSR may be driven by the ownership structure of Chinese listed firm. Whether ESOPs in other countries will have a positive effect on CSR remains to be elucidated. We thus suggest that future research may examine the effect of ESOPs on CSR under different countries’ ownership structures.

## 7. Conclusions

In June 2014, the CSRC promulgated the ‘Guidance’ to improve the governance structure of Chinese listed firms, which mainly targets employees outside of management and provides incentives to employees by associating income with corporate performance. The academic research on ESOPs in China has identified the advantages of ESOPs, such as reducing by the conflict of interest between large shareholders and employees, improving employee productivity and enhancing corporate performance [4,64]. Therefore, ESOPs have received widespread attention from practitioners and researchers. Unlike the existing literature, our paper focuses on the relationship between ESOPs and CSR. Using a sample of 895 A-share listed firms in China, we found that ESOPs promote CSR by motivating employees’ long-term orientation and mutual monitoring.

As ownership structures are crucial to the effectiveness of ESOPs, we further examined the impact of wedge structure on the relationship between ESOPs and CSR. Previous studies have found that the wedge structure facilitates controlling shareholders to expropriate the interests of minority shareholders [65,66]. Our study found that the separation of control and cash flow rights is detrimental to the effectiveness of ESOPs. Specifically, ESOPs have a stronger effect on CSR in firms without wedge structure. We also determined the impact of firm size on the relationship between ESOPs and CSR. Specifically, we found that the free-rider effect persists in CSR and that the effect of ESOPs on CSR is stronger in small firms, where mutual monitoring and group cooperation are more likely to occur.

In summary, ESOPs in Chinese listed firms provide effective incentives for employees and help firms achieve CSR goals. In particular, the incentive effect of ESOPs is stronger in firms without wedge structure, i.e., smaller firms.

## Figures and Tables

**Figure 1 ijerph-19-01055-f001:**
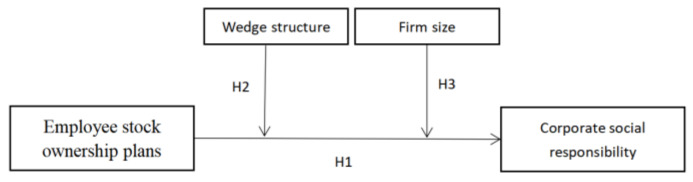
Research framework.

**Table 1 ijerph-19-01055-t001:** CSR indicator definitions.

Level 1	Level 2	Indicator Definition
Philanthropy and volunteerism	Philanthropy	1 if the firm has charitable donations; otherwise 0.
	Support education	1 if the company has behaviors that support education, such as starting a school, donating money to the Hope Project, and subsidizing poor students; otherwise 0.
	Support charity	1 if the firm has projects to support charitable donations, such as establishing its own charitable fund or cooperating with other groups to promote charitable causes; otherwise 0.
	Volunteer activities	1 if the firm has outstanding volunteer activities; otherwise 0.
	International assistance	1 if the firm engages in acts of assistance to foreign countries; otherwise 0.
	Employment generation	1 if the firm has policies or measures to promote employment which are implemented accordingly; otherwise 0.
	Promoting the local economy	1 if the corporate operations promote the economic development of local communities and if corporate policies and measures promote local economic development such as localized procurement policies and localized employment policies; otherwise 0.
	Other advantages	1 if firms make a contribution to society not mentioned in the above indicators; otherwise 0.
CSR disclosure	CSR report comprehensiveness	If the coverage of CSR information is comprehensive: 1 if the CSR report covers shareholders, creditors, employees, customers, communities and the environment or if it explicitly states that it uses the G3 standard preparation system; otherwise 0.
	CSR column	1 if there is a CSR column on the firm homepage; otherwise 0.
	CSR leading organization	1 if the firm has established a CSR leadership organization or has a clear CSR authority; otherwise 0.
	CSR vision	1 if the firm has an economically, socially and environmentally responsible philosophy, vision or values; otherwise 0.
	CSR training	1 if the firm has CSR training; otherwise 0.
	Reliability guarantee	1 if the CSR report has a reliability guarantee; otherwise 0.
	Other advantages	1 if the firm has other advantages related to CSR reporting that are not mentioned in the above indicators; otherwise 0.
Employee relations	Employee Benefits	1 if the firm has a very good retirement and other benefits program; otherwise 0.
	Safety management system	1 if the firm has a safety management system; otherwise 0.
	Safety production training	1 if the firm has conducted safety production training; otherwise 0.
	Occupational safety certification	1 if the firm is certified in occupational safety; otherwise 0.
	Vocational training	1 if the firm has vocational training for its employees; otherwise 0.
	Employee communication channels	1 if the firm has good communication channels for employee opinions or suggestions to reach the top; otherwise 0.
	Other advantages	1 if the firm has other advantages in corporate employee relations not mentioned in the above indicators; otherwise 0.
	Environmentally beneficial products	1 if the firm has developed or applied an innovative product, equipment or technology that is beneficial to the environment; otherwise 0.
Environmental responsibility	Measures to reduce triple wastes	1 if the firm has adopted policies, measures or technologies to reduce emissions of waste gases, waste water, sludge and greenhouse gases; otherwise 0.
	Circular economy	1 if the firm has policies or practices of using renewable energy or adopting a circular economy; otherwise 0.
	Energy saving	1 if the firm has policies, measures or technologies to save energy; otherwise 0.
	Green office	1 if the firm has green office policies or practice; otherwise 0.
	Environmental certification	1 if the firm’s environmental management system is ISO 14001 certified; otherwise 0.
	Environmental recognition	1 if the firm has received an environmental award or other positive evaluation; otherwise 0.
	Other advantages	1 if the firm has other advantages in corporate environment that are not mentioned in the above indicators; otherwise 0.
	Quality system	1 if the firm has product quality management systems; otherwise 0.
Product	After-sales service	1 if the firm continues to improve its after-sales services; otherwise 0.
	Customer satisfaction survey	1 if the firm has conducted customer satisfaction surveys; otherwise 0.
	Quality Accolades	1 if the firm has received certifications and awards for product quality; otherwise 0.
	Anti-corruption measures	1 if the firm has anti-commercial bribery measures or anti-corruption measures; otherwise 0.
	Strategy sharing	1 if the firm and business partners have established strategic sharing mechanisms and platforms, including long-term strategic cooperation agreements, shared experimental bases, shared databases and stable communication platforms; otherwise 0.
	Integrity management philosophy	1 if the firm has the concept and institutional guarantee of honest operation and fair competition; otherwise 0.
	Other advantages	1 if the firm has other advantages in product that are not mentioned in the above indicators; otherwise 0.

**Table 2 ijerph-19-01055-t002:** Descriptive statistics of variables.

Variable	N	Mean	Median	S.D.	Min	Max
ESOP	3550	0.123	0	0.329	0	1
CSR_FRA	3550	2.849	2.890	0.326	1.099	3.611
CSR_LOG	3550	2.271	2.232	0.735	0.250	4.750
AGE	3550	13.14	14	6.664	0	28
SIZE	3550	23.25	23.13	1.451	18.49	28.52
ROA	3550	0.041	0.035	0.052	−0.164	0.202
LEV	3550	0.482	0.491	0.197	0.0770	0.892
MANAGE	3550	0.089	0.074	0.076	0.001	0.907
CASH	3550	0.159	0.133	0.108	0.003	0.783
DUALITY	3550	1.776	1	0.462	0	1
INDRATE	3550	0.377	0.364	0.058	0.231	0.800
GTOWN	3550	0.134	0.085	0.212	−0.237	1.122
INCOME	3550	0.147	0.089	0.343	−0.471	2.140
WEDGE	3550	0.401	0	0.490	0	1
FIRMSIZE	3550	0.717	1	0.450	0	1

Note: Table 2 reports the descriptive statistics for the key variables.

**Table 3 ijerph-19-01055-t003:** Employee stock ownership plans and CSR.

	CSR_FRA			CSR_LOG		
Variables	(1)	(2)	(3)	(4)	(5)	(6)
ESOP	0.036 **	0.039 **	0.041 ***	0.086 **	0.094 ***	0.099 ***
	(2.369)	(2.572)	(2.706)	(2.563)	(2.779)	(2.971)
WDEGE		0.001			0.001	
		(0.558)			(0.152)	
FIRMSIZE			−0.078 ***			−0.187 ***
			(−5.788)			(−6.271)
ESOP*WEDGE		−0.005 ***			−0.011 ***	
		(−2.814)			(−2.798)	
ESOP*FIRMSIZE			−0.057 *			−0.192 ***
			(−1.712)			(−2.595)
SIZE	0.095 ***	0.095 ***	0.079 ***	0.243 ***	0.243 ***	0.202 ***
	(21.713)	(21.732)	(15.570)	(24.931)	(24.942)	(18.036)
ROA	0.365 ***	0.353 ***	0.342 ***	0.761 ***	0.742 ***	0.701 ***
	(3.181)	(3.073)	(3.003)	(2.995)	(2.915)	(2.781)
LEV	−0.033	−0.034	−0.022	−0.160 **	−0.162 **	−0.132 *
	(−0.932)	(−0.967)	(−0.619)	(−2.025)	(−2.051)	(−1.690)
CASH	−0.003 ***	−0.003 ***	−0.002 ***	−0.005 ***	−0.005 ***	−0.005 ***
	(−3.393)	(−3.307)	(−3.124)	(−3.130)	(−2.999)	(−2.859)
AGE	−0.058 **	−0.058 **	−0.055 **	−0.098*	−0.097 *	−0.090
	(−2.262)	(−2.231)	(−2.142)	(−1.713)	(−1.688)	(−1.592)
MANAGE	−0.016	−0.016	−0.016	−0.032	−0.033	−0.034
	(−1.034)	(−1.055)	(−1.069)	(−0.958)	(−0.973)	(−1.016)
GROWN	−0.226 ***	−0.235 ***	−0.257 ***	−0.197	−0.217	−0.271 *
	(−3.087)	(−3.212)	(−3.527)	(−1.212)	(−1.339)	(−1.676)
INCOME	−0.168 *	−0.170 **	−0.166 *	−0.244	−0.254	−0.238
	(−1.959)	(−1.979)	(−1.954)	(−1.287)	(−1.339)	(−1.268)
INDRATE	0.004	0.005	0.004	0.013	0.016	0.013
	(0.387)	(0.512)	(0.401)	(0.548)	(0.676)	(0.563)
Year	Yes	Yes	Yes	Yes	Yes	Yes
Industry	Yes	Yes	Yes	Yes	Yes	Yes
Constant	0.696 ***	0.693 ***	1.131 ***	−3.289 ***	−3.289 ***	−2.221 ***
	(7.023)	(6.991)	(9.467)	(−14.989)	(−14.977)	(−8.405)
N	3550	3550	3550	3550	3550	3550
R2	0.233	0.235	0.243	0.261	0.262	0.273
Adjust_R2	0.227	0.229	0.237	0.255	0.256	0.267
F	38.27	36.05	37.71	44.32	41.73	44.09

Note: Table 3 reports the regression results for our hypotheses. The results are from OLS regression. The t-statistics reported in parentheses are based on standard errors clustered by firm. * *p* < 0.1. ** *p* < 0.05. *** *p* < 0.01.

**Table 4 ijerph-19-01055-t004:** Results of lagging effect regression CSR.

**Panel A.** Lag one period of CSR.						
	**CSR_FRAt + 1**			**CSR_LOGt + 1**		
Variables	(1)	(2)	(3)	(4)	(5)	(6)
ESOP	0.044 **	0.049 **	0.051 ***	0.118 ***	0.131 ***	0.138 ***
	(2.287)	(2.529)	(2.695)	(2.745)	(3.017)	(3.232)
WEDGE		0.001			0.001	
		(0.701)			(0.287)	
FIRMSIZE			−0.087 ***			−0.215 ***
			(−5.664)			(−6.244)
ESOP*WEDGE		−0.006 ***			−0.014 ***	
		(−2.795)			(−2.885)	
ESOP*FIRMSIZE			−0.087 ***			−0.215 ***
			(−5.664)			(−6.244)
SIZE	0.096 ***	0.096 ***	0.077 ***	0.250 ***	0.250 ***	0.201 ***
	(18.722)	(18.766)	(12.961)	(21.629)	(21.662)	(15.124)
ROA	0.388 ***	0.365 **	0.369 ***	0.870 ***	0.827 **	0.822 ***
	(2.722)	(2.554)	(2.612)	(2.709)	(2.568)	(2.588)
LEV	−0.023	−0.025	−0.009	−0.135	−0.137	−0.097
	(−0.554)	(−0.590)	(−0.210)	(−1.421)	(−1.444)	(−1.038)
CASH	−0.003 ***	−0.003 ***	−0.003 ***	−0.006 ***	−0.005 **	−0.005 **
	(−3.085)	(−3.020)	(−2.833)	(−2.655)	(−2.542)	(−2.400)
AGE	−0.013	−0.013	−0.011	−0.014	−0.013	−0.010
	(−0.463)	(−0.435)	(−0.396)	(−0.218)	(−0.195)	(−0.155)
MANAGE	−0.010	−0.011	−0.011	−0.032	−0.033	−0.034
	(−0.575)	(−0.594)	(−0.599)	(−0.812)	(−0.831)	(−0.859)
GROWN	−0.216 **	−0.225 **	−0.251 ***	−0.153	−0.173	−0.237
	(−2.439)	(−2.537)	(−2.852)	(−0.766)	(−0.866)	(−1.199)
INCOME	−0.187 *	−0.191 *	−0.184 *	−0.310	−0.325	−0.298
	(−1.839)	(−1.867)	(−1.816)	(−1.348)	(−1.412)	(−1.311)
INDRATE	0.008	0.010	0.008	0.011	0.015	0.010
	(0.633)	(0.764)	(0.606)	(0.384)	(0.520)	(0.353)
Year	Yes	Yes	Yes	Yes	Yes	Yes
Industry	Yes	Yes	Yes	Yes	Yes	Yes
Constant	0.651 ***	0.645 ***	1.155 ***	−3.495 ***	−3.501 ***	−2.225 ***
	(5.636)	(5.571)	(8.271)	(−13.424)	(−13.430)	(−7.093)
N	2630	2630	2630	2630	2630	2630
R2	0.234	0.237	0.248	0.261	0.264	0.279
Adjust_R2	0.226	0.228	0.239	0.254	0.256	0.271
F	29.48	27.78	29.52	34.10	32.14	34.62
**Panel B.** Lag two periods of CSR.						
	**CSR_FRAt\+ 2**			**CSR_LOGt + 2**		
**Variables**	**(1)**	**(2)**	**(3)**	**(4)**	**(5)**	**(6)**
ESOP	0.062 **	0.067 ***	0.070 ***	0.170 ***	0.183 ***	0.194 ***
	(2.481)	(2.669)	(2.841)	(2.962)	(3.168)	(3.412)
WEDGE		0.001			0.001	
		(0.601)			(0.365)	
FIRMSIZE			−0.088 ***			−0.233 ***
			(−4.902)			(−5.608)
ESOP*WEDGE		−0.006 **			−0.015 **	
		(−2.240)			(−2.298)	
ESOP*FIRMSIZE			−0.090 *			−0.311 **
			(−1.675)			(−2.498)
SIZE	0.100 ***	0.100 ***	0.081 ***	0.267 ***	0.267 ***	0.214 ***
	(16.400)	(16.420)	(11.389)	(18.908)	(18.913)	(13.146)
ROA	0.436 ***	0.411 **	0.413 **	0.926 **	0.873 **	0.864 **
	(2.595)	(2.433)	(2.476)	(2.382)	(2.237)	(2.248)
LEV	−0.027	−0.029	−0.014	−0.146	−0.147	−0.108
	(−0.551)	(−0.574)	(−0.278)	(−1.263)	(−1.275)	(−0.950)
CASH	−0.003 **	−0.003**	−0.002 *	−0.005 *	−0.005 *	−0.004
	(−2.250)	(−2.206)	(−1.901)	(−1.888)	(−1.818)	(−1.504)
AGE	−0.017	−0.016	−0.011	−0.025	−0.024	−0.008
	(−0.493)	(−0.467)	(−0.311)	(−0.317)	(−0.296)	(−0.099)
MANAGE	−0.004	−0.004	−0.004	−0.018	−0.019	−0.021
	(−0.158)	(−0.171)	(−0.200)	(−0.354)	(−0.374)	(−0.420)
GROWN	−0.114	−0.118	−0.159	0.060	0.051	−0.055
	(−1.074)	(−1.108)	(−1.494)	(0.244)	(0.208)	(−0.224)
INCOME	−0.197	−0.204 *	−0.190	−0.328	−0.349	−0.307
	(−1.632)	(−1.684)	(−1.585)	(−1.175)	(−1.248)	(−1.113)
INDRATE	−0.008	−0.006	−0.007	−0.024	−0.018	−0.022
	(−0.518)	(−0.365)	(−0.467)	(−0.657)	(−0.496)	(−0.604)
Year	Yes	Yes	Yes	Yes	Yes	Yes
Industry	Yes	Yes	Yes	Yes	Yes	Yes
Constant	0.643 ***	0.635 ***	1.148 ***	−3.666 ***	−3.677 ***	−2.311 ***
	(4.661)	(4.593)	(6.886)	(−11.500)	(−11.498)	(−6.019)
N	1883	1883	1883	1883	1883	1883
R2	0.223	0.225	0.236	0.250	0.252	0.267
Adjust_R2	0.213	0.214	0.225	0.240	0.241	0.257
F	21.33	19.96	21.20	24.75	23.16	25.08

Note: Table 4 reports the regression results of lagging effect regression CSR. The results are from OLS regression. The t-statistics reported in parentheses are based on standard errors clustered by firm. * *p* < 0.1. ** *p* < 0.05. *** *p* < 0.01.

**Table 5 ijerph-19-01055-t005:** Results of difference-in-difference tests.

	CSR_FRA			CSR_LOG		
Variables	(1)	(2)	(3)	(4)	(5)	(6)
AFTER*TREAT	0.054 **	0.057 **	0.064 ***	0.119 **	0.126 **	0.143 ***
	(2.219)	(2.339)	(2.614)	(2.201)	(2.318)	(2.658)
TREAT	−0.020	−0.019	−0.024	−0.035	−0.034	−0.047
	(−0.953)	(−0.941)	(−1.189)	(−0.776)	(−0.750)	(−1.036)
WEDGE		0.001			0.001	
		(0.587)			(0.174)	
FIRMSIZE			−0.078 ***			−0.188 ***
			(−5.828)			(−6.304)
AFTER*TREAT*WEDGE		−0.005 ***			−0.011 ***	

AFTER*TREAT*FIRMSIZE			−0.057 *			−0.192 ***
			(−1.705)			(−2.589)
SIZE	0.095 ***	0.095 ***	0.078 ***	0.242 ***	0.242 ***	0.201 ***
	(21.674)	(21.694)	(15.502)	(24.895)	(24.907)	(17.971)
ROA	0.370 ***	0.358 ***	0.348 ***	0.770 ***	0.750 ***	0.712 ***
	(3.219)	(3.109)	(3.051)	(3.025)	(2.943)	(2.822)
LEV	−0.032	−0.033	−0.020	−0.157 **	−0.159 **	−0.128
	(−0.887)	(−0.924)	(−0.562)	(−1.988)	(−2.015)	(−1.639)
AGE	−0.003 ***	−0.003 ***	−0.003 ***	−0.006 ***	−0.005 ***	−0.005 ***
	(−3.471)	(−3.386)	(−3.224)	(−3.192)	(−3.059)	(−2.946)
GROWN	−0.058 **	−0.057 **	−0.054 **	−0.097 *	−0.095 *	−0.089
	(−2.236)	(−2.205)	(−2.109)	(−1.692)	(−1.667)	(−1.564)
INCOME	−0.016	−0.016	−0.016	−0.032	−0.033	−0.034
	(−1.021)	(−1.043)	(−1.053)	(−0.948)	(−0.964)	(−1.002)
MANAGE	−0.224 ***	−0.233 ***	−0.255 ***	−0.192	−0.213	−0.266
	(−3.054)	(−3.179)	(−3.490)	(−1.186)	(−1.314)	(−1.644)
INDRATE	−0.171 **	−0.173 **	−0.170 **	−0.250	−0.260	−0.247
	(−1.997)	(−2.015)	(−2.003)	(−1.319)	(−1.367)	(−1.311)
DUALITY	0.004	0.005	0.004	0.013	0.016	0.013
	(0.363)	(0.488)	(0.371)	(0.528)	(0.657)	(0.537)
Year	Yes	Yes	Yes	Yes	Yes	Yes
Industry	Yes	Yes	Yes	Yes	Yes	Yes
Constant	0.703 ***	0.700 ***	1.143 ***	−3.276 ***	−3.277 ***	−2.197 ***
	(7.076)	(7.042)	(9.534)	(−14.884)	(−14.877)	(−8.288)
N	3550	3550	3550	3550	3550	3550
R2	0.234	0.235	0.244	0.261	0.263	0.273
Adjust_R2	0.227	0.229	0.237	0.255	0.256	0.267
F	36.98	34.92	36.55	42.81	40.40	42.71

Note: Table 5 reports the regression results of the DID test. The t-statistics reported in parentheses are based on standard errors clustered by firm. * *p* < 0.1. ** *p* < 0.05. *** *p* < 0.01.

**Table 6 ijerph-19-01055-t006:** Results of Heckman two-stage tests.

Variables	ESOP	CSR_FRA			CSR_LOG		
	(1)	(2)	(3)	(4)	(5)	(6)	(7)
ESOP		0.036 **	0.039 **	0.041 ***	0.085 **	0.093 ***	0.098 ***
		(2.334)	(2.542)	(2.669)	(2.516)	(2.738)	(2.922)
WEDGE			0.000			0.000	
			(0.466)			(0.050)	
FIRMSIZE				−0.081 ***			−0.194 ***
				(−5.979)			(−6.479)
ESOP*WEDGE			−0.005 ***			−0.011 ***	
			(−2.755)			(−2.734)	
ESOP*FIRMSIZE				−0.055 *			−0.187 **
				(−1.649)			(−2.526)
SIZE	0.125 **	0.099 ***	0.098 ***	0.083 ***	0.252 ***	0.248 ***	0.213 ***
	(2.560)	(5.644)	(5.544)	(4.733)	(6.472)	(6.367)	(5.467)
ROA	−1.478	0.330	0.340	0.294	0.661	0.691	0.569
	(−1.189)	(1.430)	(1.474)	(1.282)	(1.292)	(1.352)	(1.122)
LEV	0.287	−0.017	−0.022	−0.002	−0.121	−0.132	−0.085
	(0.715)	(−0.317)	(−0.416)	(−0.035)	(−1.036)	(−1.131)	(−0.735)
CASH	−0.039 ***	−0.004	−0.003	−0.004	−0.009	−0.007	−0.009
	(−4.634)	(−0.761)	(−0.646)	(−0.796)	(−0.750)	(−0.624)	(−0.789)
AGE	0.861 ***	−0.030	−0.041	−0.018	−0.030	−0.056	−0.002
	(3.505)	(−0.257)	(−0.353)	(−0.157)	(−0.115)	(−0.215)	(−0.009)
MANAGE	0.398 ***	−0.002	−0.008	0.001	0.000	−0.013	0.008
	(2.683)	(−0.040)	(−0.147)	(0.021)	(0.003)	(−0.105)	(0.062)
GROWN	2.968 ***	−0.117	−0.168	−0.120	0.067	−0.049	0.061
	(4.340)	(−0.292)	(−0.418)	(−0.299)	(0.075)	(−0.055)	(0.070)
INCOME	−2.226 **	−0.237	−0.208	−0.257	−0.412	−0.350	−0.456
	(−2.260)	(−0.751)	(−0.658)	(−0.820)	(−0.588)	(−0.500)	(−0.658)
INDRATE	−0.267 **	−0.005	−0.000	−0.008	−0.009	0.003	−0.015
	(−2.467)	(−0.137)	(−0.001)	(−0.208)	(−0.104)	(0.036)	(−0.179)
Lambda		0.039	0.023	0.050	0.094	0.057	0.120
		(0.252)	(0.146)	(0.326)	(0.274)	(0.165)	(0.353)
Year	Yes	Yes	Yes	Yes	Yes	Yes	Yes
Industry	Yes	Yes	Yes	Yes	Yes	Yes	Yes
Constant	−5.791 ***	0.473	0.565	0.855	−3.833 *	−3.616 *	−2.886
	(−5.154)	(0.524)	(0.626)	(0.952)	(−1.917)	(−1.809)	(−1.453)
N	3515	3515	3515	3515	3515	3515	3515
R2	0.1022	0.232	0.234	0.243	0.260	0.261	0.273
Adjust_R2	.	0.226	0.228	0.237	0.254	0.255	0.267
F	.	39.08	36.72	38.55	45.28	42.53	45.11

Note: Table 6 reports the results of the Heckman two-stage model. The t-statistics reported in parentheses are based on standard errors clustered by firm. * *p* < 0.1. ** *p* < 0.05. *** *p* < 0.01.

## Data Availability

The data will be made available on request.

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
