# Peer review of "Do Employee Stock Ownership Plans Affect Corporate Social Responsibility? Evidence from China"

_ijerph, 2022, doi:10.3390/ijerph19031055_

Round 1

Reviewer 1 Report

This is a well developed paper that deserves publication. I would however suggest to authors to consider the following:

Please provide additional explanation for the claim on p3, line 101: „After 2007, the internal employee share dropped to 0.“

P3, line 105 correct technical error

P4 – the relationship between CSR and agency theory should be supported with appropriate references. Agency theory is used to explain management-shareholder relationship but mostly does not include employees as such. Please add some references that explain employee behaviour through agency theory.

P9 – can you please add short explanation why size is measured by total asset and not with number of employees, as you previously explained free-rider problem

In conclusion, you should emphasize more theoretical and practical implications of your work, as well as research limitations

Author Response

Thank you for your review our manuscript. Those comments are valuable and very helpful. We have read through comments carefully and have made corrections. 

Comment 1:Please provide additional explanation for the claim on p3, line 101: “After 2007, the internal employee share dropped to 0”.

Response:We have added a more detailed interpretation regarding “internal employee share dropped to 0”. More detailed explanation was added on page p3, line 102 and 103. Owing to the lack of effective supervision, a large number of irregularities exist in the ESOPs at this stage, resulting in a significant loss of state-owned asset, the Chinese government eventually terminated the internal employee share ownership policy. Since then, the employee shareholding of Chinese listed companies has gradually decreased, and after 2007, all employee shareholdings have been converted into tradable shares.

Comment 2P3, line 105 correct technical error.

Response:We apologize for the technical error in the original manuscript. We have made relevant corrections. Please see Page 3, line 110.

Comment 3P4–the relationship between CSR and agency theory should be supported with appropriate references. Agency theory is used to explain management-shareholder relationship but mostly does not include employees as such. Please add some references that explain employee behaviour through agency theory.

Response:We agree with the comment and add some references to support our view in the revised manuscript as the following: Prior literature argue that managers and workers are natural allies against takeover threats because takeovers and subsequent mergers are often associated with layoffs (Pagano and Volpin, 2004; Aubert, 2014). Please see Page 3, line 105. In fact, our manuscript explains the agency conflict between employee and shareholder at fulfilling CSR. Similar views about agency conflict between employee and shareholder are also reflected in the study of Meng et al. (2019) et al.  

Comment 4Can you please add short explanation why size is measured by total asset and not with number of employees, as you previously explained free-rider problem.

Response:We agree with the comment and add explanation in the revised manuscript. we use total assets rather than number of employees to measure firm size is largely attribute to the fact that totals asset are an important factor affecting a firm's CSR investment. Please see Page 9, line 327.

Comment 5you should emphasize more theoretical and practical implications of your work, as well as research limitations

Response:Thank you for your suggestion. As suggested by reviewer, we have added the discussion part in the revised manuscript. Please see Page 17 and 18.

References

Pagano, M. Volpin, P F. Managers, Workers, and Corporate Control[J]. Journal of Fiance. 2005, 60(2): 841-868.

Aubert N, Garnotel G, Lapied A, et al. Employee ownership: A theoretical and empirical investigation of management entrenchment vs. reward management[J]. Economic Modelling, 2014, 40(6):423-434.

Meng Qingbin, Li Xinyu, Zhang Peng. Can Employee Stock Ownership Plan Promote Enterprise Innovation?——Empirical Evidence Based on the Perspective of Enterprise Employees[J]. Management World, 2019, 35(11): 209-228.

Reviewer 2 Report

This is a very interesting topic! I encourage you to consider the following recommendations to improve your work.

1-Please explain the method and the software that you used in the abstract and conclusion sections.

2-In the abstract you mentioned, “This research provides insights for understanding the relationship between ESOPs and non-financial performance and has important managerial implications for firms to pay attention to the interests of employees to achieve sustainable development.” This needs to be addressed in more detail before the conclusion section.

3-Please convince your readers how you assessed the reliability of your adopted method.

4-Right after reporting the results you jumped into the conclusion section. Please add the discussion and limitations section. For this study, it's important to highlight all of the limitations in detail. In the discussion part, you need to address some of the possible implications of your study.

5-I encourage you to take a look at the following studies, there are valuable information in these papers related to your work. you might find them valuable for your current or future studies: Please consider citing Chiu and Tsai , Ahmed, and one of Block and Wagner papers to your study.

Block, J., & Wagner, M. (2014). Ownership versus management effects on corporate social responsibility concerns in large family and founder firms. Journal of Family Business Strategy5(4), 339-346.

Block, J. H., & Wagner, M. (2014). The effect of family ownership on different dimensions of corporate social responsibility: Evidence from large US firms. Business Strategy and the Environment23(7), 475-492.

Chiu, S. F., & Tsai, W. C. (2007). The linkage between profit sharing and organizational citizenship behaviour. The International Journal of Human Resource Management18(6), 1098-1115.

Von Arx, U., & Ziegler, A. (2008). The effect of CSR on stock performance: new evidence for the USA and Europe. CER-ETH-Center of Economic Research at ETH Zurich, Working Paper, (08/85).

Ahmed, D. B. (2020). Behavioral effects of employee stock ownership: French case. Asian Journal of Empirical Research10(2), 53-64.

Kim, A., & Han, K. (2019). All for one and one for all: A mechanism through which broad‐based employee stock ownership and employee‐perceived involvement practice create a productive workforce. Human Resource Management58(6), 571-584.

Author Response

Thank you for your review our manuscript. Those comments are valuable and very helpful. We have read through comments carefully and have made corrections. Based on your comment, we revise our manuscript.

Comment 1: Please explain the method and the software that you used in the abstract and conclusion sections.

Response:We are grateful for the suggestion. We have added the method and software used in this article to the abstract. Our research mainly used the OLS model to test the research hypotheses, and all regressions were performed in Stata15. Please see Page 1, line 11.

Comment 2: In the abstract you mentioned, “This research provides insights for understanding the relationship between ESOPs and non-financial performance and has important managerial implications for firms to pay attention to the interests of employees to achieve sustainable development.” This needs to be addressed in more detail before the conclusion section.

Response:Accordance with the reviewer suggestion, we have added a more detailed about this view in the discussion part. Please see page 18.

Comment 3: Please convince your readers how you assessed the reliability of your adopted method.

Response : In fact, to ensure the robustness of the results, we use the alternative CSR measure and CSR lag term for robustness testing. At the same time, we also use the difference-in-differences model and Heckman two-stage model for endogeneity testing to ensure that our research does not suffer from serious endogeneity problems. We have added relevant instructions in the revised manuscript. Please see Page 10, line 351-355.

Comment 4: Right after reporting the results you jumped into the conclusion section. Please add the discussion and limitations section. For this study, it's important to highlight all of the limitations in detail. In the discussion part, you need to address some of the possible implications of your study.

Response : Thank you for your suggestion. As suggested by reviewer, we have added the discussion part and possible implications in the revised manuscript. Please see Page 17 and 18.

Comment 5: I encourage you to take a look at the following studies, there are valuable information in these papers related to your work. you might find them valuable for your current or future studies: Please consider citing Chiu and Tsai , Ahmed, and one of Block and Wagner papers to your study.  

Response:  As suggested by reviewer, we cite Block and Wagner, and Chiu and Tsai papers in our study. Please see Page 1, line 44 and 45; Page 5, line 213.

Reviewer 3 Report

Interesting article describing important issues. The article is well covered with literature, hypotheses and arguments clearly and legibly presented (albeit in line 233 H1 instead of "Compared firms without ESOPs, firms with ESOPs have higher CSR" it would be better to use the provision "Compared to firms without ESOPs, firms with ESOPs have higher CSR" ). The presented research model is understandable, and the research methodology is well described. The presented research results are convincing, although their number may slightly overwhelm the reader. Summing up, in terms of content, a very good study on an interesting and relatively rarely discussed issue. I congratulate the authors on the results of their hard work.

In terms of "technical" issues, I generally recommend that you check the content of the article for grammatical compliance and language style - preferably by a native speaker.

In line 105/106 - Error! Bookmark not defined 

Author Response

Thank you for your review our manuscript. Those comments are valuable and very helpful. We have read through comments carefully and have made corrections. Based on your comment, we revise our manuscript. 

Comment 1: Albeit in line 233 H1 instead of "Compared firms without ESOPs, firms with ESOPs have higher CSR" it would be better to use the provision "Compared to firms without ESOPs, firms with ESOPs have higher CSR"

Response : Thank you for your suggestion. We have revised our manuscript. Please see Page 5, line 241.

Comment 2: In line 105/106 - Error! Bookmark not defined.

Response: We apologize for the technical error in the original manuscript. We have made relevant corrections. Please see Page 3, line 110.

Round 2

Reviewer 2 Report

Thank you for addressing all of the concerns. Well done!